# Nationwide mammography screening participation in Denmark during the COVID-19 pandemic: An observational study

**Tina Bech Olesen[1]\*, Henry Jensen[1], Henrik Møller[1], Jens Winther Jensen[1], Berit Andersen[2,3], Ilse Vejborg[4], Sisse H Njor[1,2,3]**

[1]The Danish Clinical Quality Program – National Clinical Registries (RKKP), Aarhus N, Denmark; [2]University Research Clinic for Cancer Screening, Department of Public Health Programmes, Randers Regional Hospital, Randers, Denmark; [3]Department of Clinical Medicine, Aarhus University, Aarhus, Denmark; [4]Department of Breast Examinations, Copenhagen University Hospital, Copenhagen, Denmark

**\*For correspondence:**
forthemanuscripts@gmail.com

**Competing interest:** The authors declare that no competing interests exist.

## Abstract

**Background:** In most of the world, the mammography screening programmes were paused at the start of the pandemic, whilst mammography screening continued in Denmark. We examined the mammography screening participation during the COVID-19 pandemic in Denmark.

**Methods:** The study population comprised all women aged 50–69 years old invited to participate in mammography screening from 2016 to 2021 in Denmark based on data from the Danish Quality Database for Mammography Screening in combination with population-based registries. Using a generalised linear model, we estimated prevalence ratios (PRs) and 95% confidence intervals (CIs) of mammography screening participation within 90, 180, and 365 d since invitation during the pandemic in comparison with the previous years adjusting for age, year and month of invitation.

**Results:** The study comprised 1,828,791 invitations among 847,766 women. Before the pandemic, 80.2% of invitations resulted in participation in mammography screening within 90 d, 82.7% within 180 d, and 83.1% within 365 d. At the start of the pandemic, the participation in screening within 90 d was reduced to 69.9% for those invited in pre-lockdown and to 76.5% for those invited in first lockdown. Extending the length of follow-up time to 365 d only a minor overall reduction was observed (PR = 0.94; 95% CI: 0.93–0.95 in pre-lockdown and PR = 0.97; 95% CI: 0.96–0.97 in first lockdown). A lower participation was, however, seen among immigrants and among women with a low income.

**Conclusions:** The short-term participation in mammography screening was reduced at the start of the pandemic, whilst only a minor reduction in the overall participation was observed with longer follow-up time, indicating that women postponed screening. Some groups of women, nonetheless, had a lower participation, indicating that the social inequity in screening participation was exacerbated during the pandemic.

**Funding:** The study was funded by the Danish Cancer Society Scientific Committee (grant number R321-A17417) and the Danish regions.

## Editor's evaluation

This article is of broad interest to public health researchers and health policymakers in populations with national screening programmes. It provides important knowledge on the impact of the

COVID-19 pandemic on participation in mammography screening in Denmark by socio-economic indicators. The study provides convincing evidence for how the pandemic exacerbated disparities in breast cancer screening in Denmark.

## Introduction

The COVID-19 pandemic has abruptly impacted the healthcare system and the society at large. Population-wide restrictions ('lockdowns') were imposed worldwide to minimise the spread of infection with COVID-19 and to lessen the burden on the healthcare systems. Within some healthcare systems, resources were reallocated to ensure sufficient capacity to take care of COVID-19 patients in need of hospitalisation.

The prioritisations within the healthcare system resulted in a temporary closure of the cancer screening programmes in most countries worldwide. The reason for this was mainly to decrease the risk of women being infected by COVID-19 and due to the shortage of health professionals as they were involved in pandemic-related activities. Congruently, the European Society for Medical Oncology deemed population-based mammography screening to be of low priority at the start of the pandemic (*de Azambuja et al., 2020*). Nevertheless, in Denmark the mammography screening programme continued – and invitations and reminders were sent out – throughout the pandemic since early detection of cancer was considered of high priority. Studies from other countries have shown marked reductions in breast cancer screening during the pandemic as a result of the temporary closures of the screening programmes (*Ng and Hamilton, 2022*; *Cairns et al., 2022*; *Jidkova et al., 2022*; *Miller et al., 2021*; *Sprague et al., 2021*); however, the participation in mammography screening in Denmark with a continuation of the programme has not yet been described.

Disruptions to the mammography screening programme could result in later diagnosis of breast cancer and a more advanced stage of diagnosis (*Figueroa et al., 2021*), and the disruptions to the programme are therefore worrisome. Congruently, studies from the Netherlands found that the temporary closure of the screening programme resulted in a reduction in the number of breast cancer diagnoses, in particular screen-detected breast cancers (*Eijkelboom et al., 2021b*; *Dinmohamed et al., 2020*) and in particular among early-stage tumours (*Eijkelboom et al., 2021a*). In Denmark, an early study reported that a 19% reduction in the number of breast cancers was observed in the spring of 2020 (*Skovlund et al., 2022*). Contrarily, more recent data from the Danish Breast Cancer Group show a modest reduction of 5% in the number of breast cancer in 2020 compared to 2019 (*The Danish Clinical Quality Program – National Clinical Registries, 2022*).

The participation in mammography screening is generally lower among women with low socio-economic status and among immigrant women (*Bhargava et al., 2018*; *Kristiansen et al., 2012*; *Jensen et al., 2012*). This gap may have worsened during the pandemic. One study from Spain *Bosch et al., 2022* found that the participation in mammography screening decreased with age and with lower socio-economic level post-COVID-19; however, to our knowledge, no studies have examined participation according to ethnicity, cohabitation status, or healthcare use throughout the pandemic.

We examined the participation in mammography screening in Denmark during the COVID-19 pandemic in comparison with the previous years. Moreover, we assessed whether the participation in mammography screening during the COVID-19 pandemic differed according to socio-economic variables.

## Methods
### Setting

The study is set in Denmark, which has a population of approximately 5.8 million inhabitants (*Statistics Denmark, 2021*). In Denmark, there is free access to healthcare for all inhabitants funded via taxes. Danish population-based administrative and health registries can be linked using the unique personal identifier, assigned to all residents at birth or immigration (*Schmidt et al., 2014*; *Schmidt et al., 2019*).

### The mammography screening programme

In Denmark, all women aged 50-69 years old are invited to participate in mammography screening every 2 y (except women, who have opted out of the programme). The mammography screening

programme is administered by the five Danish regions. Digital invitation letters with a fixed appointment and link to an information leaflet are sent to the women. In the capital region, previous non-participants who have not responded to neither invitation nor reminder letter in the previous screening round do not receive a fixed appointment but only an invitation letter when it is time for the next mammography screening. The women can reschedule their appointments by e-mail, through a website, or by telephone. At the mammography screening session, two standardised X-ray images of each breast are performed and these images are evaluated independently by two radiologists (*Mikkelsen et al., 2016*).

## The COVID-19 pandemic in Denmark

Three main waves of the COVID-19 pandemic have occurred in Denmark, that is, in the spring of 2020, in the winter of 2020/2021, and again in the winter of 2021/2022 (*Statens Serum Institut, 2021a*).

The pandemic response in Denmark comprised 'lockdowns,' COVID-19 testing, and COVID-19 vaccination. The first lockdown was imposed in Denmark on 11 March 2020, closing down schools and workplaces and restricting travel. This was done in an effort to minimise the spread of infection and reduce the potential burden on the healthcare system. Within the healthcare system, elective procedures were cancelled or postponed and resources were reallocated to take care of patients in need of hospitalisation because of COVID-19. The subsequent lockdowns occurred together with the new surges of infection. Widespread COVID-19 testing was implemented in Denmark from May 2020 providing PCR and antigen COVID-19 tests free of charge to the whole population (*Pottegård et al., 2020*). Vaccination against COVID-19 was implemented in Denmark in December 2020, and by March 2022, approximately 81% of the population had received two doses and more than 61% had received three doses of the vaccine (*Statens Serum Institut, 2021b*). In Denmark, individuals living in nursing homes were vaccinated first, followed by individuals ≥85 y, then healthcare personnel, thereafter individuals with underlying health conditions and their relatives, and finally, individuals were offered the COVID-19 vaccination by decreasing age (75–79 y, 65–74 y, 60–64 y, etc.) (*Sundhedsstyrelsen, 2021*).

## Study population

We included all invitations to women aged 50–69 y invited to participate in mammography screening from 1 January 2016 to 30 September 2021 as registered in the Danish Quality Database for Mammography Screening (*Mikkelsen et al., 2016*), which retrieves information on invitations to the national mammography screening programme directly from the regional administrative systems.

We excluded invitations to women with an unknown postal address (N = 753 in 406 women), invitations to women residing in the Faroe Islands or Greenland (N = 310 in 220 women), invitations to women with unknown/uncertain living status (N = 199 in 146 women), invitations to women who emigrated within 1 y since invitation (N = 2692 in 2,581 women), invitations to women who died within 1 y since invitation (N = 8860 in 8,854 women), and women excluded due to dates being out of order (e.g. pre-booked date for mammography screening was prior to date of invitation; N = 8218 in 1,023 women) (*Figure 1*).

## Exposure of interest

The COVID-19 pandemic in Denmark and the different phases of the pandemic are the exposure of interest. The different phases of the pandemic were defined in accordance with the governmental responses to the COVID-19 pandemic in Denmark as follows:

- Pre-pandemic period: 1 January 2016 to 31 January 2020
- Pre-lockdown period: 1st February to 10 March 2020
- First lockdown: 11 March to 15 April 2020
- Second re-opening: 16 April to 15 December 2020
- Second lockdown: 16 December 2020 to 27 February 2021
- Second reopening: 28 February 2021 to 30 September 2021 (end of inclusion period)

Pre-lockdown and first lockdown were considered the start of the pandemic in this study.

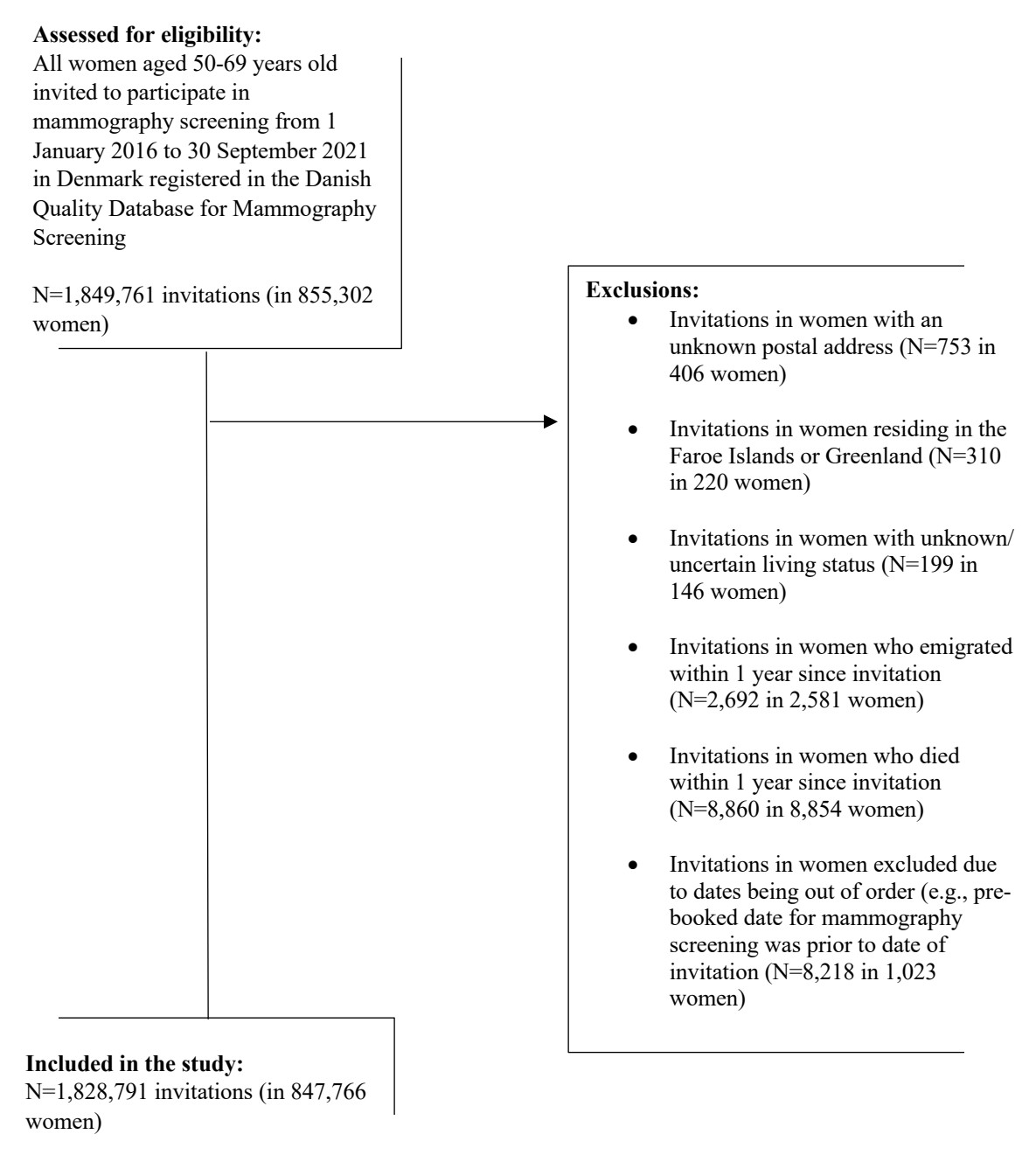

**Assessed for eligibility:**
All women aged 50-69 years old invited to participate in mammography screening from 1 January 2016 to 30 September 2021 in Denmark registered in the Danish Quality Database for Mammography Screening

N=1,849,761 invitations (in 855,302 women)

**Exclusions:**
- Invitations in women with an unknown postal address (N=753 in 406 women)
- Invitations in women residing in the Faroe Islands or Greenland (N=310 in 220 women)
- Invitations in women with unknown/ uncertain living status (N=199 in 146 women)
- Invitations in women who emigrated within 1 year since invitation (N=2,692 in 2,581 women)
- Invitations in women who died within 1 year since invitation (N=8,860 in 8,854 women)
- Invitations in women excluded due to dates being out of order (e.g., pre-booked date for mammography screening was prior to date of invitation (N=8,218 in 1,023 women)

**Included in the study:**
N=1,828,791 invitations (in 847,766 women)

**Figure 1.** Flowchart of the study population.

## Outcome of interest

The main outcome of interest was participation in mammography screening defined as women who underwent a mammography screening within 90, 180, and 365 d, respectively, among women invited to participate in mammography screening in each of the defined periods.

## Explanatory variables

We examined the following variables independently: age, ethnicity, cohabitation status, educational level, disposable income, and healthcare usage. Age was defined as the date of invitation, as registered in the Danish Quality Database for Mammography Screening (*Mikkelsen et al., 2016*). From *Statistics Denmark, 2021*, we obtained information on ethnicity, cohabitation status, educational

level, and disposable income. Ethnicity was categorised as Danish descent, Western immigrant, and non-Western immigrant and descendants of immigrants in accordance with *Statistics Denmark, 2021*. Cohabitation status was categorised as living alone, co-habiting/co-living, and married (i.e. married or registered partnership) in accordance with *Statistics Denmark, 2021*. Education level was defined in accordance with the International Standard Classification of Education (ISCED) of the United Nations Education, Scientific and Cultural Organization into short (ISCED levels 1-2), medium (ISCED levels 3–4), and long (ISCED levels 5–8) (*Statistics Denmark, 2021*). Income was defined as official disposable income depreciated to 2015 level and categorised into five quintiles. To indicate the level of healthcare use by each patient, we counted the total number of contacts to general practitioners, private practising medical specialists, physiotherapists, and chiropractors in the year for invitation as registered in the Danish National Health Service Register (*Andersen et al., 2011*), which contains information on visits to primary healthcare (e.g. general practitioners and medical specialists) in Denmark since 1990. We categorised healthcare usage as rare (0–3 visits per year), low (4–6 visits per year), average (7–11 visits per year), high (12–18 visits per year), and frequent (≥19 visits per year).

## Statistical analyses

Firstly, we examined the baseline characteristics of women invited to participate in mammography screening during the study period. Subsequently, we examined the proportion of women participating in mammography screening within 90, 180, and 365 d since invitation per month and during the different phases of the pandemic overall and stratifying by the explanatory variables.

Thereafter, we estimated prevalence ratios (PRs) and 95% confidence intervals (CIs) of mammography screening participation within 90, 180, and 365 d since invitation during the different phases of the pandemic overall and stratifying by the explanatory variables using a generalised linear model with robust standard error. Initially, unadjusted analyses were performed. Thereafter, the analyses were adjusted for month of invitation to take into account seasonality and year of invitation to allow for the annual change in participation. Finally, the analyses were adjusted for age to take into account the effect of age on the other explanatory variables.

All analyses were conducted using STATA version 17.0.

## Ethical considerations

The study is registered at the Central Denmark Region's register of research projects (journal number 1-16-02-381-20). According to Danish law, register-based studies should not be reported to the National Committee on Health Research Ethics. Furthermore, patient consent is not required by Danish law for register-based studies.

## Results

### Descriptive characteristics of the study population

Altogether, 847,766 women receiving 1,828,791 invitations were included in the study. The majority were of Danish descent (91.7%), the median age was 58 years old (interdecentile interval [IQI] = 54–64), most were married (60.2%), and the majority had a short educational level (58.7%). The distribution of the descriptive characteristics was largely similar across the different phases of the pandemic (*Table 1*).

### Participation during the COVID-19 pandemic

Before the pandemic, 80.2% of women participated in mammography screening within 90 d since invitation, 82.7% within 180 d, and 83.1% within 365 d (*Figure 2* and *Supplementary files 1–3*).

At the start of the pandemic, a reduction of 3.7–10.6 percentage points in screening participation within 90 d was observed (PR = 0.85; 95% CI: 084–0.85 in pre-lockdown and PR = 0.93; 95% CI: 0.92–0.94 in first lockdown) corresponding to a participation rate of 69.6% among women invited during pre-lockdown and 76.5% among women invited during the first lockdown, respectively. Thereafter the participation within 90 d resumed to the same level as before the pandemic (*Supplementary files 1 and 4*).

The participation in mammography screening within 365 d was also reduced by 1.6–3.5 percentage points (PR = 0.94; 95% CI: 0.93–0.95 among women invited during pre-lockdown and PR = 0.97;

**Table 1.** Baseline characteristics of women invited to participate in mammography screening in Denmark from 2016 to 2021.

| | Pre-pandemic (1 January 2016 to 31 January 2020) | Pre-lockdown (1 February 2020 to 10 March 2020) | First lockdown (11 March 2020 to 15 April 2020) | First reopening (16 April 2020 to 15 December 2020) | Second lockdown (16 December 2020 to 27 February 2021) | Second reopening (28 February 2021 to 30 September 2021) | Total |
|---|---|---|---|---|---|---|---|
| | N (%) | N (%) | N (%) | N (%) | N (%) | N (%) | N (%) |
| Total | 1,346,959 (73.7) | 35,175 (1.9) | 14,134 (0.8) | 201,613 (11.0) | 61,991 (3.4) | 168,919 (9.2) | 1,828,791 (100.0) |
| *Age at invitation (y)* | | | | | | | |
| 50–54 | 421,813 (31.3) | 10,660 (30.3) | 3998 (28.3) | 66,325 (32.9) | 19,792 (31.9) | 48,669 (28.8) | 57,1257 (31.2) |
| 55–59 | 326,444 (24.2) | 9439 (26.8) | 3516 (24.9) | 49,481 (24.5) | 14,914 (24.1) | 44,000 (26.0) | 447,794 (24.5) |
| 60–64 | 305,389 (22.7) | 7824 (22.2) | 3362 (23.8) | 44,793 (22.2) | 13,897 (22.4) | 38,487 (22.8) | 413,752 (22.6) |
| 65–69 | 293,313 (21.8) | 7252 (20.6) | 3258 (23.1) | 41,014 (20.3) | 13,388 (21.6) | 37,763 (22.4) | 395,988 (21.7) |
| Median age (IQI) | 58 (54; 64) | 58 (54; 64) | 59 (54; 64) | 58 (54; 64) | 58 (54; 64) | 58 (54; 64) | 58 (54; 64) |
| *Ethnicity* | | | | | | | |
| Danish descent | 1,237,422 (91.9) | 32,030 (91.1) | 12,811 (90.6) | 182,860 (90.7) | 49,328 (91.4) | 136,255 (92.2) | 1,650,706 (91.7) |
| Descendant of immigrant | 2213 (0.2) | 52 (0.1) | 26 (0.2) | 358 (0.2) | 70 (0.1) | 237 (0.2) | 2956 (0.2) |
| Western immigrant | 35,894 (2.7) | 1042 (3.0) | 456 (3.2) | 5823 (2.9) | 1436 (2.7) | 3659 (2.5) | 48,310 (2.7) |
| Non-Western immigrant | 71,293 (5.3) | 2047 (5.8) | 841 (6.0) | 12,557 (6.2) | 3124 (5.8) | 7572 (5.1) | 97,434 (5.4) |
| *Cohabitation status* | | | | | | | |
| Living alone | 411,431 (30.6) | 11,161 (31.7) | 4504 (31.9) | 61,551 (30.5) | 2362 (30.4) | N/A | 491,009 (30.6) |
| Cohabiting | 122,870 (9.1) | 3373 (9.6) | 1301 (9.2) | 19,823 (9.8) | 720 (9.3) | N/A | 148,087 (9.2) |
| Married | 812,442 (60.3) | 20,632 (58.7) | 8327 (58.9) | 120,209 (59.6) | 4678 (60.3) | N/A | 966,288 (60.2) |
| *Educational level (ISCED)* | | | | | | | |
| ISCED15 levels 1–2 | 766,480 (57.8) | 21,103 (60.8) | 8453 (60.6) | 121,233 (60.9) | 37,204 (60.9) | 103,826 (62.4) | 1,058,299 (58.7) |
| ISCED15 levels 3–4 | 397,781 (30.0) | 9642 (27.8) | 3904 (28.0) | 54,964 (27.6) | 16,609 (27.2) | 44,245 (26.6) | 527,145 (29.2) |
| ISCED15 levels 5–8 | 162,943 (12.3) | 3991 (11.5) | 1588 (11.4) | 22,755 (11.4) | 7316 (12.0) | 18,406 (11.1) | 216,999 (12.0) |
| *Disposable income* | | | | | | | |
| Lowest quintile | 271,215 (20.1) | 6306 (17.9) | 2642 (18.7) | 34,833 (17.3) | 10,176 (16.4) | 27,802 (16.5) | 352,974 (19.3) |
| Second quintile | 270,148 (20.1) | 6849 (19.5) | 2790 (19.7) | 37,553 (18.7) | 11,000 (17.8) | 30,887 (18.3) | 359,227 (19.7) |
| Third quintile | 273,315 (20.3) | 7088 (20.2) | 2757 (19.5) | 40,063 (19.9) | 11,220 (18.1) | 30,898 (18.3) | 365,341 (20.0) |
| Fourth quintile | 269,796 (20.0) | 7126 (20.3) | 2840 (20.1) | 43,042 (21.4) | 13,115 (21.2) | 35,539 (21.1) | 371,458 (20.3) |
| Highest quintile | 262,055 (19.5) | 7768 (22.1) | 3102 (22.0) | 45,789 (22.7) | 16,440 (26.5) | 43,540 (25.8) | 378,694 (20.7) |
| *Healthcare usage* | | | | | | | |
| Rare | 301,815 (22.4) | 7894 (22.4) | 3070 (21.7) | 45,216 (22.4) | 13,344 (21.5) | 37,009 (21.9) | 408,348 (22.3) |
| Low | 240,671 (17.9) | 6145 (17.5) | 2473 (17.5) | 35,618 (17.7) | 10,607 (17.1) | 29,030 (17.2) | 324,544 (17.7) |
| Average | 292,576 (21.7) | 7667 (21.8) | 3122 (22.1) | 43,970 (21.8) | 13,526 (21.8) | 36,837 (21.8) | 397,698 (21.7) |
| High | 248,125 (18.4) | 6607 (18.8) | 2693 (19.1) | 37,863 (18.8) | 11,962 (19.3) | 32,309 (19.1) | 339,559 (18.6) |
| Frequent | 263,772 (19.6) | 6862 (19.5) | 2776 (19.6) | 38,946 (19.3) | 12,552 (20.2) | 33,734 (20.0) | 358,642 (19.6) |

*Table 1 continued on next page*

*Table 1 continued*

| | Pre-pandemic (1 January 2016 to 31 January 2020) | Pre-lockdown (1 February 2020 to 10 March 2020) | First lockdown (11 March 2020 to 15 April 2020) | First reopening (16 April 2020 to 15 December 2020) | Second lockdown (16 December 2020 to 27 February 2021) | Second reopening (28 February 2021 to 30 September 2021) | Total |
|---|---|---|---|---|---|---|---|
| | N (%) | N (%) | N (%) | N (%) | N (%) | N (%) | N (%) |

IQR = interquartile range. IQI = interdecentile interval (10, 50, 90). ISCED = International Standard Classification of Education

95% CI: 0.96–0.97 among women invited during the first lockdown) corresponding to a participation rate of 79.6% during pre-lockdown and 81.5% during the first lockdown. Again, the overall participation resumed to pre-pandemic levels from first reopening and onwards (*Table 2* and *Supplementary file 3*).

The results for participation within 180 d since invitation were similar to the results for participation within 365 d since invitation (*Figure 2* and *Supplementary files 2 and 5*).

## Participation during the COVID-19 pandemic according to socio-economic variables

Before the pandemic, the participation in mammography screening within 90 and 365 d was the lowest among the youngest women (77.6 and 80.9%), among immigrants (67.8 and 70.7% among Western and 66.8 and 69.9% among non-Western immigrants), among women living alone (72.1 and 75.2%), among women with the lowest income (73.8 and 76.1%), and among women who rarely use the primary healthcare system (75.4 and 78.2%) (*Supplementary files 1-3*).

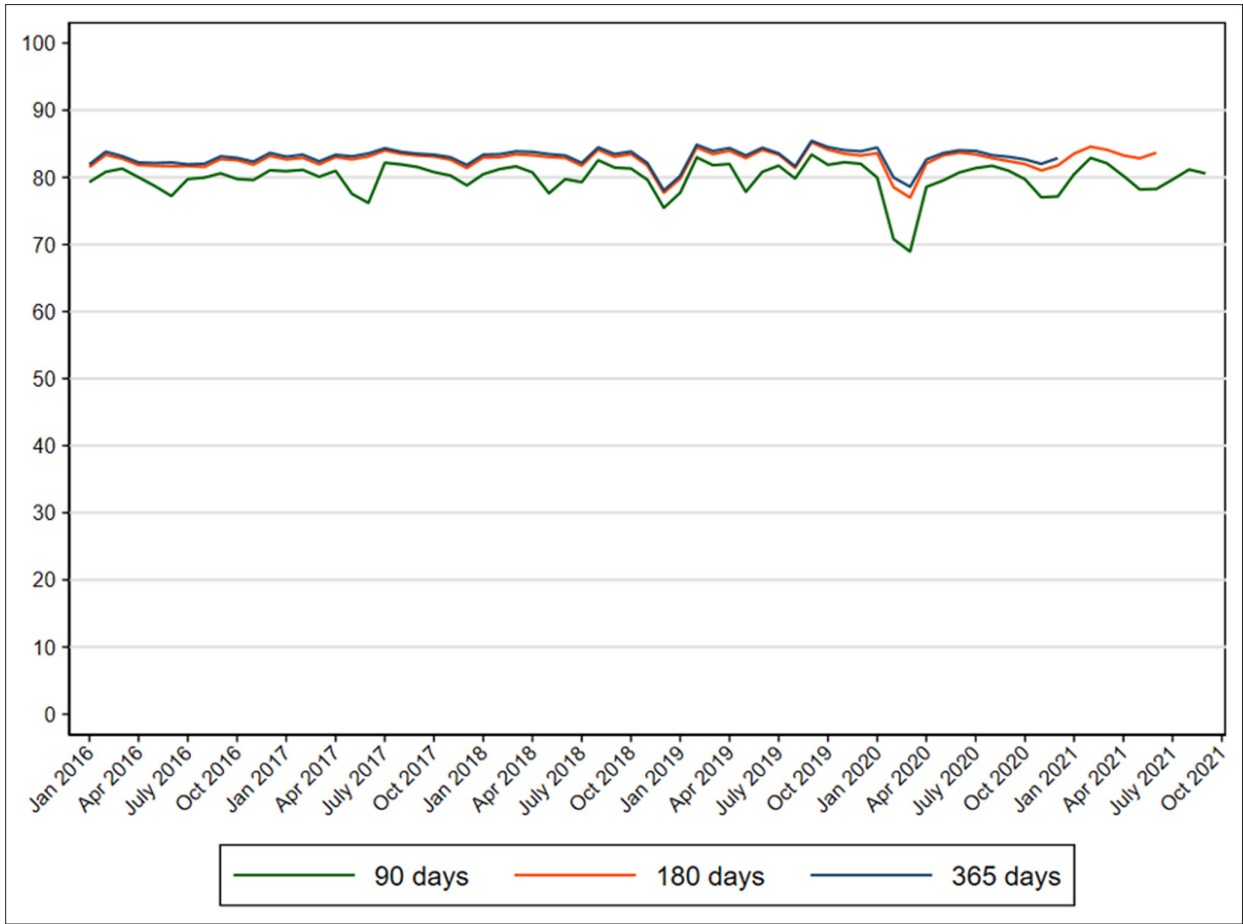

**Figure 2.** Proportion of women participating in mammography screening in Denmark within 90, 180, and 365 d since invitation from 2016 to 2021.

**Table 2.** Prevalence ratios (PR) and 95% confidence intervals (95% CI) of participation in mammography screening in Denmark within 365 d since invitation from 2016 to 2021*.

| | | Pre-pandemic (1 January 2016 to 31 January 2020) | | Pre-lockdown (1 February 2020 to 10 March 2020) | | First lockdown (11 March 2020 to 15 April 2020) | | First reopening (16 April 2020 to 15 December 2020) | | Second lockdown (16 December 2020 to 31 December 2020) | |
|---|---|---|---|---|---|---|---|---|---|---|---|
| | | N = 1,346,959 | | N = 35,175 | | N = 14,134 | | N = 201,613 | | N = 61,991 | |
| | N | PR | [95% CI] | PR | [95% CI] | PR | [95% CI] | PR | [95% CI] | PR | [95% CI] |
| Overall | 1,828,791 | 1.00 | - | 0.94 | [0.93; 0.95] | 0.97 | [0.96; 0.97] | 0.99 | [0.99; 0.99] | 1.00 | [0.99; 1.01] |
| *Age at invitation (y)* | | | | | | | | | | | |
| 50–54 | 571,257 | 1.00 | - | 0.94 | [0.93; 0.95] | 0.96 | [0.95; 0.98] | 0.99 | [0.99; 1.00] | 0.97 | [0.95; 0.99] |
| 55–59 | 447,794 | 1.00 | - | 0.94 | [0.93; 0.95] | 0.97 | [0.96; 0.99] | 0.99 | [0.99; 1.00] | 1.02 | [1.00; 1.04] |
| 60–64 | 413,752 | 1.00 | - | 0.94 | [0.93; 0.95] | 0.97 | [0.95; 0.98] | 0.99 | [0.99; 1.00] | 1.02 | [1.00; 1.04] |
| 65–69 | 395,988 | 1.00 | - | 0.94 | [0.93; 0.95] | 0.96 | [0.95; 0.98] | 0.98 | [0.98; 0.99] | 0.99 | [0.97; 1.01] |
| *Ethnicity* | | | | | | | | | | | |
| Danish descent | 165,070 | 1.00 | - | 0.94 | [0.94; 0.95] | 0.97 | [0.96; 0.98] | 0.99 | [0.99; 1.00] | 1.00 | [0.99; 1.01] |
| Descendant of immigrant | 2956 | 1.00 | - | 1.00 | [0.86; 1.16] | 1.09 | [0.89; 1.34] | 0.91 | [0.84; 0.98] | 1.22 | [1.10; 1.36] |
| Western immigrant | 48,310 | 1.00 | - | 0.91 | [0.87; 0.96] | 0.96 | [0.90; 1.03] | 0.97 | [0.95; 0.99] | 0.97 | [0.88; 1.06] |
| Non-Western immigrant | 97,434 | 1.00 | - | 0.88 | [0.85; 0.91] | 0.95 | [0.90; 1.00] | 0.96 | [0.95; 0.98] | 0.95 | [0.89; 1.02] |
| *Cohabitation status* | | | | | | | | | | | |
| Living alone | 491,009 | 1.00 | - | 0.92 | [0.91; 0.93] | 0.95 | [0.94; 0.97] | 0.98 | [0.97; 0.98] | 0.98 | [0.95; 1.00] |
| Cohabiting | 148,087 | 1.00 | - | 0.92 | [0.91; 0.94] | 0.96 | [0.93; 0.99] | 0.99 | [0.98; 1.00] | 1.02 | [0.98; 1.05] |
| Married | 966,288 | 1.00 | - | 0.95 | [0.95; 0.96] | 0.97 | [0.96; 0.98] | 0.99 | [0.99; 1.00] | 1.00 | [0.99; 1.01] |
| *Educational level (ISCED)* | | | | | | | | | | | |
| ISCED15 levels 1–2 | 105,829 | 1.00 | - | 0.94 | [0.93; 0.95] | 0.96 | [0.95; 0.97] | 0.99 | [0.99; 0.99] | 1.00 | [0.99; 1.01] |
| ISCED15 levels 3–5 | 527,145 | 1.00 | - | 0.94 | [0.93; 0.95] | 0.97 | [0.96; 0.98] | 0.99 | [0.99; 1.00] | 1.00 | [0.98; 1.02] |
| ISCED15 levels 6–8 | 216,999 | 1.00 | - | 0.94 | [0.92; 0.96] | 0.98 | [0.96; 1.01] | 0.99 | [0.98; 1.00] | 1.01 | [0.98; 1.04] |
| *Disposable income* | | | | | | | | | | | |
| Lowest quintile | 352,974 | 1.00 | - | 0.91 | [0.89; 0.92] | 0.94 | [0.91; 0.96] | 0.98 | [0.97; 0.99] | 0.95 | [0.91; 0.98] |
| Second quintile | 359,227 | 1.00 | - | 0.92 | [0.90; 0.93] | 0.95 | [0.93; 0.97] | 0.97 | [0.96; 0.98] | 0.99 | [0.96; 1.02] |
| Third quintile | 365,341 | 1.00 | - | 0.95 | [0.93; 0.96] | 0.96 | [0.95; 0.98] | 1.00 | [0.99; 1.00] | 1.00 | [0.98; 1.02] |
| Fourth quintile | 371,458 | 1.00 | - | 0.95 | [0.94; 0.96] | 0.99 | [0.97; 1.00] | 1.00 | [0.99; 1.00] | 1.01 | [0.99; 1.03] |
| Highest quintile | 378,694 | 1.00 | - | 0.97 | [0.96; 0.98] | 0.99 | [0.97; 1.00] | 0.99 | [0.99; 1.00] | 1.02 | [1.00; 1.03] |
| *Healthcare usage* | | | | | | | | | | | |
| Rare | 408,348 | 1.00 | - | 0.93 | [0.92; 0.95] | 0.96 | [0.94; 0.98] | 0.98 | [0.98; 0.99] | 0.99 | [0.97; 1.02] |
| Low | 324,544 | 1.00 | - | 0.95 | [0.94; 0.96] | 0.97 | [0.95; 0.99] | 0.99 | [0.98; 1.00] | 1.01 | [0.99; 1.03] |
| Average | 397,698 | 1.00 | - | 0.94 | [0.93; 0.95] | 0.97 | [0.95; 0.99] | 0.99 | [0.99; 1.00] | 0.99 | [0.96; 1.01] |
| High | 339,559 | 1.00 | - | 0.95 | [0.93; 0.96] | 0.95 | [0.94; 0.97] | 1.00 | [0.99; 1.00] | 1.00 | [0.98; 1.03] |
| Frequent | 358,642 | 1.00 | - | 0.94 | [0.93; 0.95] | 0.98 | [0.96; 0.99] | 0.99 | [0.98; 0.99] | 1.00 | [0.98; 1.03] |

ISCED = International Standard Classification of Education.

*Adjusted for month, year and age at invitation.

The reduction in participation within 90 d at the start of the pandemic was most pronounced in the oldest age group (to 70.3% in pre-lockdown and 76.6% in first lockdown), among immigrants (to 55.1% among Western and 53.2% among non-Western immigrants in pre-lockdown), among women with a low income (to 60.1% in pre-lockdown and 66.7% in first lockdown), and women who frequently

use the primary healthcare system (to 68.6% in pre-lockdown). A slightly lower participation remained among women with a low income throughout the study period (*Supplementary files 1 and 4*).

Extending the length of follow-up time to 365 d, a lower participation remained among immigrants (65.6% among Western and 64.2% among non-Western immigrants in pre-lockdown) and among women with a low income (69.3% in pre-lockdown and 71.5% in first lockdown) for those invited at the start of the pandemic. The reduced participation was observed throughout the study period among women with a low income (*Table 2* and *Supplementary file 3*).

The overall as well as the results stratifying by socio-economic variables were similar in both the unadjusted model and when adjusting for month and any underlying decreasing/increasing trends (data not shown).

## Discussion

### Main findings

In this nationwide population-based study comprising 1,828,791 invitations among 847,766 women, we found a reduction in participation in mammography within 90 d at the start of the pandemic; however, extending the length of follow-up time to 365 d, only a minor reduction was observed. From first reopening and onwards, the participation in mammography screening normalised. A lower participation was, nevertheless, seen among immigrants and among women with a low income.

### Comparison with previous studies and explanation of findings

Throughout most of the world, the mammography screening programmes were paused at the start of the pandemic. The screening programmes were stopped mainly to decrease the risk of women being infected by COVID-19 while being screened, but also due to the shortage of health professionals, as they were involved in pandemic-related activities (e.g. radiologist addressed to emergency department to assure diagnosis of pneumonia). This led to marked reductions in screening mammographies performed at the start of the pandemic in many other countries worldwide (*Ng and Hamilton, 2022*; *Cairns et al., 2022*; *Jidkova et al., 2022*; *Miller et al., 2021*; *Sprague et al., 2021*). In Denmark, the mammography screening programme continued throughout the pandemic, and the situation in Denmark is therefore unique and few studies are comparable to our study.

We found a short-term reduction in participation in mammography screening at the start of the pandemic, which may have been caused by the governmental instructions to stay at home. Congruently, a qualitative study from Denmark found that women weighed the benefits of screening versus the risk of acquiring COVID-19 in their decision-making process as to whether to participate in mammography screening during the pandemic (*Kirkegaard et al., 2021*). This was also demonstrated in a study from the United States (*Schifferdecker et al., 2021*). With longer follow-up time, we found that most women did participate in screening, indicating a postponement of participation in mammography screening. COVID-19 vaccination was implemented in Denmark in December 2020, which could explain the unaltered participation in mammography screening during the second reopening of the society. During the pandemic, initiatives were made throughout Denmark to inform women of the possibility of participating in mammography screening during the pandemic and to inform women about safety measures when participating in screening; for example, a letter was sent to women opting out of the programme (i.e. withdrawing from the screening programme) in some regions, in some regions press releases were sent out and in some regions, and women calling in to cancel their appointment for mammography screening were individually counselled and encouraged to consider participation later on (instead of waiting for the next biannual round of screening). This could explain the overall modest reduction in participation. Before the pandemic, 83.1% of women participated in mammography screening within 365 d, and this was reduced to 79.6% for those invited during pre-lockdown and to 81.5% for those invited during the first lockdown, indicating that a minor reduction in participation remained even with the longest follow-up time. However, the participation in screening for cervical and bowel cancer in Denmark during the pandemic showed similar patterns, with a decrease in the short term, but to a much lesser extent in the longer term. This indicates that people invited for screening in Denmark more often postponed their participation by a few months, rather than not attending at all (*Olesen et al., 2023a*; *Olesen et al., 2023b*; this study).

The decreased participation at the start of the pandemic was most pronounced among immigrants and in particular among women with a low income. The participation dropped by 3.3% among women of Danish descent for those invited during pre-lockdown, whilst the participation dropped by 5.0% among Western and 5.7% among non-Western immigrants. Even larger differences were seen when stratifying by income level: among women with the lowest income, the participation dropped by 6.9% during pre-lockdown and by 4.6% during the first lockdown, whilst the participation among women with the highest income was almost unchanged. This indicates that the restrictions imposed in Denmark may have affected some groups of women disproportionally. A study from Spain *Bosch et al., 2022* found that the participation in mammography screening decreased with age and with lower socio-economic level post-COVID-19; this has also been reported for the two other screening programmes in Denmark (*Olesen et al., 2023a*; *Olesen et al., 2023b*; this study). The evidence is scarce on participation according to ethnicity throughout the pandemic; however, the lower participation during the pandemic has also been demonstrated for bowel and cervical cancer screening in Denmark (*Olesen et al., 2023a*; *Olesen et al., 2023b*; this study). Immigrants may find it more difficult to navigate the healthcare system in Denmark, and possibly this has been exacerbated by the pandemic where the majority of the communication at press conferences and news media were conveyed in Danish, a language that may not be fully comprehensive to all immigrants. Moreover, the COVID-19 vaccination coverage was lower among immigrants (*Gram et al., 2023*), which could partly explain the lower screening participation. Women with a low income may not have access to private means of transport and may have been reluctant to use public transportation to reach a mammography screening clinic during the pandemic, which could explain the lower participation among this group of women. Women without private means of transport generally have a lower participation in mammography screening (*Jensen et al., 2012*; *Jensen et al., 2014*), and this may have worsened during the pandemic. The lower participation in screening continued throughout the study period among women with a low income, indicating that the pandemic had lasting negative effects on the participation in mammography screening in this group of women.

## Strengths and limitations

An important strength of the study is the high quality of data covering the entire population of women invited to participate in the mammography screening programme in Denmark during the pandemic and the years before. Danish administrative and health registries have a high completeness (*Thygesen et al., 2011*), which is also the case for the Danish Quality Database for Mammography Screening (*Mikkelsen et al., 2016*). Limitations of the study should, however, be acknowledged. In this study, we calculated participation in mammography screening from the date of invitation to the date of participation; however, the length of time from invitation to the pre-booked appointment may vary depending on the capacity of the mammography screening clinics. A more accurate measure may have been time from first pre-booked appointment to participation; however, this information is unfortunately not available in the Danish Quality Database for Mammography Screening. We did not have data on underlying disease or vaccination status, which could affect one's perception of risk and willingness to participate in screening. We did adjust for age, and since age is strongly associated with the level of comorbidity, the theoretical impact of comorbidity is reduced. Moreover, the severity of the pandemic and the pandemic response was different in different countries, with Denmark managing to keep the number of hospitalisations due to COVID-19 at a relatively low level (*Statens Serum Institut, 2021a*), and our observations may not be directly transferable to all countries.

## Implications of the findings

Our study indicates that women postponed screening at the start of the pandemic. The mammography screening remained open throughout the pandemic, and this is contrasting to the health communication conveyed at the national televised press conferences at the start of the pandemic in Denmark instructing people to stay at home and cancel appointments. Inconsistent health messages may thus have resulted in a reduced participation in screening. The pandemic was an unprecedented situation, and the health authorities had to navigate unknown terrain; however, in hindsight it is important to ensure that the health communication is aligned among all parties to ensure that women both feel safe and that it is indeed safe to participate in screening.

We identified groups of women who had a low participation in screening before the pandemic and found that for some groups the participation was even lower at the start of the pandemic (immigrants and women with a low income), and for women with a low income the participation remained low throughout the pandemic. Our study thus indicates that the social inequity in mammography screening participation was slightly exacerbated during the pandemic.

The effect of the short-term reduction in mammography screening participation on the detection of early-stage breast cancer is unknown. A study from Denmark by *Skovlund et al., 2022* reported a 19% reduction in breast cancers diagnosed in the spring of 2020; however, this large reduction may partly be a result of a delayed registration of cancer diagnoses at the time of that study as cancer diagnoses have been shown to be incompletely registered for months (*Larsen et al., 2014*). Furthermore, more recent data from the Danish Breast Cancer Group show a modest reduction of 5% in the number of breast cancers in 2020 compared to 2019 (*The Danish Clinical Quality Program – National Clinical Registries, 2022*). Nonetheless, some women may have changed their health-seeking behaviour possibly not contacting their general practitioner at the start of the pandemic despite symptoms of breast cancer. Moreover, our study supports that delayed participation in mammography screening might to some degree contribute to a reduction in the number of breast cancers diagnosed in the spring of 2020. However, since by far most women merely postponed their participation and showed up later, the lower participation at the start of the pandemic most likely only has a minor effect on the overall mortality from breast cancer at a population level.

## Conclusion

We found that the short-term mammography screening participation was reduced at the start of the pandemic, whilst only a minor reduction in participation was observed with longer follow-up time. Some groups of women (immigrants and women with a low income) had a lower participation even with the longest follow-up time, indicating that the social inequity in mammography screening participation was slightly exacerbated during the pandemic.

## Acknowledgements

The study was funded by the Danish Cancer Society Scientific Committee (grant number R321-A17417) and the Danish regions.

## Additional information

### Funding

| Funder | Grant reference number | Author |
| --- | --- | --- |
| The Danish Cancer Society Scientific Committee | R321-A17417 | Tina Bech Olesen |
| The Danish regions | | Tina Bech Olesen Henry Jensen |

The funders had no role in study design, data collection and interpretation, or the decision to submit the work for publication.

### Author contributions

Tina Bech Olesen, Conceptualization, Data curation, Funding acquisition, Methodology, Writing - original draft, Writing – review and editing; Henry Jensen, Conceptualization, Data curation, Formal analysis, Methodology, Writing – review and editing; Henrik Møller, Jens Winther Jensen, Conceptualization, Funding acquisition, Writing – review and editing; Berit Andersen, Conceptualization, Supervision, Funding acquisition, Writing – review and editing; Ilse Vejborg, Writing – review and editing; Sisse H Njor, Methodology, Writing – review and editing

### Author ORCIDs

Tina Bech Olesen http://orcid.org/0000-0002-6295-7399
Henry Jensen http://orcid.org/0000-0003-4040-7334

Sisse H Njor [ORCID] https://orcid.org/0000-0003-0429-4176

### Ethics

The study is registered at the Central Denmark Region's register of research projects (journal number 1-16-02-381-20). According to Danish law, register-based studies should not be reported to the National Committee on Health Research Ethics. Furthermore, patient consent is not required by Danish law for register-based studies.

### Decision letter and Author response

Decision letter https://doi.org/10.7554/eLife.83541.sa1
Author response https://doi.org/10.7554/eLife.83541.sa2

---

## Additional files

### Supplementary files

• Supplementary file 1. Proportion of women participating in mammography screening in Denmark within 90 d since invitation from 2016 to 2021.

• Supplementary file 2. Proportion of women participating in mammography screening in Denmark within 180 d since invitation from 2016 to 2021.

• Supplementary file 3. Proportion of women participating in mammography screening in Denmark within 365 d since invitation from 2016 to 2021.

• Supplementary file 4. Prevalence ratios and 95% confidence intervals of participation in mammography screening in Denmark within 90 d since invitation from 2016 to 2021.

• Supplementary file 5. Prevalence ratios and 95% confidence intervals of participation in mammography screening in Denmark within 180 d since invitation from 2016 to 2021.

• MDAR checklist

### Data availability

In order to comply with the Danish regulations on data privacy, the datasets generated and analysed during this project are not publicly available as the data are stored and maintained electronically at Statistics Denmark, where it only can be accessed by pre-approved researchers using a secure VPN remote access. Furthermore, no data at a personal level nor data not exclusively necessary for publication are allowed to be extracted from the secure data environment at Statistics Denmark. Access to the data can; however, be granted by the authors of the present study upon a reasonable scientific proposal within the boundaries of the present project and for scientific purposes only. Scientific proposals should be forwarded to Data Analyst, PhD Henry Jensen at HERJEN@rkkp.dk. All analyses were conducted using STATA version 17.0.

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
