## [Editor Report]

This article is of broad interest to public health researchers and health policymakers in populations with national screening programmes. It provides important knowledge on the impact of the COVID-19 pandemic on participation in mammography screening in Denmark by socio-economic indicators. The study provides convincing evidence for how the pandemic exacerbated disparities in breast cancer screening in Denmark.

---

## [Decision Letter]

**Decision letter after peer review:**

Thank you for submitting your article "Nation-wide mammography screening participation in Denmark during the COVID-19 pandemic: An observational study" for consideration by *eLife*. Your article has been reviewed by 2 peer reviewers, and the evaluation has been overseen by a Reviewing Editor and a Senior Editor. The following individual involved in review of your submission has agreed to reveal their identity: Paolo Giorgi Rossi (Reviewer #1).

As is customary in *eLife*, the reviewers have discussed their critiques with one another. What follows below is the Reviewing Editor's edited compilation of the essential and ancillary points provided by reviewers in their critiques and in their interaction post-review. Please submit a revised version that addresses these concerns directly. Although we expect that you will address these comments in your response letter, we also need to see the corresponding revision clearly marked in the text of the manuscript. Some of the reviewers' comments may seem to be simple queries or challenges that do not prompt revisions to the text. Please keep in mind, however, that readers may have the same perspective as the reviewers. Therefore, it is essential that you attempt to amend or expand the text to clarify the narrative accordingly.

Essential revisions:

1) Please add reasons or explanations around screening services shut-off.

2) The issue of safety for women and the lack of health professionals during pandemic waves should be explained as the rationale of stopping screening.

3) Consider shortening the description of results and including some discussion on the differences between the mammography program with results from the other Danish cancer screening programs (e.g., for colorectal or cervical cancer) reported on by the authors.

*Reviewer #1 (Recommendations for the authors):*

Background

Lines 97-98 "however, to our 97 knowledge no studies have examined participation according to ethnicity, cohabitation status or 98 healthcare use throughout the pandemic." I suggest avoiding sentences like this, it is extremely likely that they will be outdated in a few months. Furthermore, I am pretty sure that there are reports on the interaction between socioeconomic factors screening participation and the pandemic.

Methods

Line 192: check grammar.

Results

I suggest reducing the text. To better present the change during the pandemic, I suggest reporting the absolute difference as percentage points.

Lines 276 277: "The overall as well as the results stratifying by socio-economic variables were similar in both the unadjusted model and when adjusting for month and year of invitation (data not shown)." I do not understand how could you adjust for year of invitation if the exposure variable is based on calendar time.

Discussion

Lines 288-290: "Throughout most of the world, the mammography screening programmes were paused at the start of the pandemic. This led to marked reductions in breast cancer screening at the start of the pandemic in many other countries worldwide" this sentence sound tautologic, maybe the authors want to say led to a reduction in screen-detected cancers? Or mammography’s…

Lines 305-306: "e.g. a letter was sent to women opting out of the programme in some regions" it is not clear what you mean by "opting out": women not participating after invitation? Women actively responding to the invitation asking to be excluded? Not clear if the letter was sent to women opting out for the first time during the pandemic or to women who opted out also in previous invitations.

Lines 307-308: "women calling in to cancel their appointment for mammography screening" what's the difference between opting out and calling to cancel the appointment?

Lines 363-364: "it is important to ensure that the health communication is aligned among all parties to ensure that women feel safe to participate in screening" What is really important is that women are safe when attending the mammographic screening.

Among the limitations, the authors should mention the missed link between the screening program that was never stopped and the 19% decrease in breast cancer incidence. The authors only can present some hypotheses about. Actually, if the screening program monitoring system should be able to assess if there was a decrease in screen-detected cancers or not.

In general, in the discussion, I miss two of the main points that led to suspend screening programs in most countries during the pandemic: (1) protecting women from the risk of infection linked to attending a clinic during pandemic when health facilities were mostly attended by symptomatic people seeking care for Covid-19; (2) the of health professionals because they were mostly involved in covid related activities: lack of radiologists (addressed to the emergency department to assure diagnoses of pneumonia), lack of anesthesiologists (due to the expansion of intensive care), thus risking not having timely surgical treatment; lack of screening organization personal for invitations and phone calls (working on contact tracing). Lacking the rationale for suspending screening, it is not clear to the reader how the Danish program afforded these issues and was able to maintain open the program.

A question for the authors: what happened around January 2019?

---

## [Author Response]

Essential revisions:1) Please add reasons or explanations around screening services shut-off.

Thank you for this comment. In Denmark, the mammography screening programme was kept open throughout the COVID-19 pandemic – thus there were no shut-downs of the screening services (see Background section page 3, lines 76-78). However, we have added reasons for screening services to be shut down in other countries in the background section (page 3, lines 74-76) and in the Discussion section (page 9, lines 296-299)

2) The issue of safety for women and the lack of health professionals during pandemic waves should be explained as the rationale of stopping screening.

We agree. We have added this to the background and Discussion section (page 3, lines 76-78 and page 9, lines 296-299).

3) Consider shortening the description of results and including some discussion on the differences between the mammography program with results from the other Danish cancer screening programs (e.g., for colorectal or cervical cancer) reported on by the authors.

We have shortened the Results section by excluding 2 paragraphs regarding the participation within 180 days. Instead, we have stated with a single sentence, that the results for participation within 180 days were almost identical to participation within 365 days as follows (Result section page 7, lines 250-251):

"The results for participation within 180 days since invitation were similar to the results for participation within 365 days since invitation (Figure 1, Supplementary File 2 and 5)”.

Furthermore, we have added comparison with the other cancer screening programmes in Denmark to the Discussion section (page 9, lines 323-327 and page 10 lines 338-342).

Reviewer #1 (Recommendations for the authors):BackgroundLines 97-98 "however, to our 97 knowledge no studies have examined participation according to ethnicity, cohabitation status or 98 healthcare use throughout the pandemic." I suggest avoiding sentences like this, it is extremely likely that they will be outdated in a few months. Furthermore, I am pretty sure that there are reports on the interaction between socioeconomic factors screening participation and the pandemic.

We have revised this sentence.

MethodsLine 192: check grammar.

We have checked the grammar.

ResultsI suggest reducing the text.

We have shortened the text – please see our response to editors point #3.

To better present the change during the pandemic, I suggest reporting the absolute difference as percentage points.

Good point. We now report the difference in percentage points (page 7, lines 232, 244-245).

Lines 276 277: "The overall as well as the results stratifying by socio-economic variables were similar in both the unadjusted model and when adjusting for month and year of invitation (data not shown)." I do not understand how could you adjust for year of invitation if the exposure variable is based on calendar time.

We apologise for any confusion made. We attempted to state that the results were similar when accounting for any underlying annual trends in participation before the pandemic. We have rephrased the sentence as follows (page 8, lines 283-284):

"The overall as well as the results stratifying by socio-economic variables were similar in both the unadjusted model and when adjusting for month and any underlying decreasing/ increasing trends (data not shown).".

DiscussionLines 288-290: "Throughout most of the world, the mammography screening programmes were paused at the start of the pandemic. This led to marked reductions in breast cancer screening at the start of the pandemic in many other countries worldwide" this sentence sound tautologic, maybe the authors want to say led to a reduction in screen-detected cancers? Or mammography’s…

We indeed wanted to write reduction in screening mammographys performed. We have amended accordingly (page 9, lines 299-300).

Lines 305-306: "e.g. a letter was sent to women opting out of the programme in some regions" it is not clear what you mean by "opting out": women not participating after invitation? Women actively responding to the invitation asking to be excluded? Not clear if the letter was sent to women opting out for the first time during the pandemic or to women who opted out also in previous invitations.

Thank you for this comment. Opting out refers to women actively drops out of the screening programme permanently (or until the actively signs up again). These women are thus different from women who do not participate after an invitation i.e. non-participants. We have clarified this in the text by clarifying that opting out refers to withdrawing form the programme (page 9, line 320).

Lines 307-308: "women calling in to cancel their appointment for mammography screening" what's the difference between opting out and calling to cancel the appointment?

Thank you for this comment. When opting pout, women drops out permanently, whereas when cancelling the appointment wither leads to a reminder or a new invitation within two years time (as the screening programme is biannual). We have clarified this in the text (page 9, line 323).

Lines 363-364: "it is important to ensure that the health communication is aligned among all parties to ensure that women feel safe to participate in screening" What is really important is that women are safe when attending the mammographic screening.

We agree, and have amended the text accordingly (page 11, line 387).

Among the limitations, the authors should mention the missed link between the screening program that was never stopped and the 19% decrease in breast cancer incidence. The authors only can present some hypotheses about. Actually, if the screening program monitoring system should be able to assess if there was a decrease in screen-detected cancers or not.

Thank you for this comment. The 19% decrease is from a previous study (Skovlund et al) that was limited by the data sources used and thus a delay in the registration of cancer diagnoses. More recent data from the Danish Breast Cancer Group (DBCG) suggest that the decline in breast cancer diagnoses "only" was 5% in 2020 compared to 2019. We have clarified this in the text (page 3, lines 90-95 and page 11, lines 394-397).

In general, in the discussion, I miss two of the main points that led to suspend screening programs in most countries during the pandemic: (1) protecting women from the risk of infection linked to attending a clinic during pandemic when health facilities were mostly attended by symptomatic people seeking care for Covid-19; (2) the of health professionals because they were mostly involved in covid related activities: lack of radiologists (addressed to the emergency department to assure diagnoses of pneumonia), lack of anesthesiologists (due to the expansion of intensive care), thus risking not having timely surgical treatment; lack of screening organization personal for invitations and phone calls (working on contact tracing). Lacking the rationale for suspending screening, it is not clear to the reader how the Danish program afforded these issues and was able to maintain open the program.

Thank you for these points. Please see our response to the same issue in the public review.

A question for the authors: what happened around January 2019?

Thank you for this comment. We are not aware of any specific changes in the programme in January 2019. We expect that this is fluctuations in screening participation.